# Molecular Basis, Diagnostic Challenges and Therapeutic Approaches of Bartter and Gitelman Syndromes: A Primer for Clinicians

**DOI:** 10.3390/ijms222111414

**Published:** 2021-10-22

**Authors:** Laura Nuñez-Gonzalez, Noa Carrera, Miguel A. Garcia-Gonzalez

**Affiliations:** 1Grupo de Xenetica e Bioloxia do Desenvolvemento das Enfermidades Renais, Laboratorio de Nefroloxia (No. 11), Instituto de Investigacion Sanitaria de Santiago (IDIS), Complexo Hospitalario de Santiago de Compostela (CHUS), 15706 Santiago de Compostela, Spain; laura.nunez.gonzalez@rai.usc.es; 2Grupo de Medicina Xenomica, Complexo Hospitalario de Santiago de Compostela (CHUS), 15706 Santiago de Compostela, Spain; 3RedInRen (Red en Investigación Renal) RETIC (Redes Temáticas de Investigación Cooperativa en Salud), ISCIII (Instituto de Salud Carlos III), 28029 Madrid, Spain; 4Fundación Pública Galega de Medicina Xenomica—SERGAS, Complexo Hospitalario de Santiago de Compotela (CHUS), 15706 Santiago de Compostela, Spain

**Keywords:** Bartter syndrome, Gitelman syndrome, genetics, genetic diagnosis, therapeutic targets, hyponatremia, hypokalemia, hypercalciuria, hypomagnesemia

## Abstract

Gitelman and Bartter syndromes are rare inherited diseases that belong to the category of renal tubulopathies. The genes associated with these pathologies encode electrolyte transport proteins located in the nephron, particularly in the Distal Convoluted Tubule and Ascending Loop of Henle. Therefore, both syndromes are characterized by alterations in the secretion and reabsorption processes that occur in these regions. Patients suffer from deficiencies in the concentration of electrolytes in the blood and urine, which leads to different systemic consequences related to these salt-wasting processes. The main clinical features of both syndromes are hypokalemia, hypochloremia, metabolic alkalosis, hyperreninemia and hyperaldosteronism. Despite having a different molecular etiology, Gitelman and Bartter syndromes share a relevant number of clinical symptoms, and they have similar therapeutic approaches. The main basis of their treatment consists of electrolytes supplements accompanied by dietary changes. Specifically for Bartter syndrome, the use of non-steroidal anti-inflammatory drugs is also strongly supported. This review aims to address the latest diagnostic challenges and therapeutic approaches, as well as relevant recent research on the biology of the proteins involved in disease. Finally, we highlight several objectives to continue advancing in the characterization of both etiologies.

## 1. Introduction

The main function of renal tubules is the control of reabsorption and secretion of electrolytes, in order to maintain correct homeostasis [1]. There are numerous proteins involved directly or indirectly in this function. Any impairment that compromises their correct function will cause, to a greater or lesser degree, a dysregulation of homeostasis [2], thus, giving rise to different clinical manifestations (kidney tubulopathy). 

GS and BS are monogenic diseases belonging the group of inherited renal tubulopathies [2]. Gitelman (GS) and Bartter syndromes (BS) pathognomonic symptoms include hypokalemia, hypochloremic metabolic alkalosis, hyperreninism and secondary hyperaldostheronism due to volume-contraction consequences of the activation of the renin-angiotensin-aldosterone system (RAAS) [3,4]. The overall prevalence of inherited tubulopathies remains unknown [5]. For GS, the estimated prevalence varies from 1 to 10 per 40,000 individuals worldwide [6], whereas, for BS, the data is not well-defined but an annual incidence of 1 per 1,000,000 is estimated [7].

In 1962, Dr. C. Frederic described the clinical symptoms of a new disease named Bartter syndrome [8]. Since then, several studies have revealed the great genetic heterogeneity related to this syndrome and, to date, six genes have been linked to this pathology [9,10,11,12,13,14,15], subclassifying the disease according to the underlying genes (Table 1). On the other hand, Gitelman syndrome (GS) was first named as an independent clinical entity in 1969 [16]. Despite the fact that the causative gene (*SLC12A3*) was discovered in 1996 [17], Gitelman syndrome has been confused with a subtype of Bartter syndrome, leading to misdiagnoses for many years [18].

## 2. Renal Physiology of Electrolytes

One of the functions of the kidney is the maintenance and regulation of acid-base homeostasis [19]. This balance is maintained by the interregulation between glomerular filtration and tubular reabsorption [20,21], which makes the urine not only a way of eliminating toxic metabolites but also a complex milieu adapted to these processes [2]. In different segments of the nephron, 99% of the total glomerular filtration volume is reabsorbed via transcellular (active or passive transporters) or paracellular mechanisms [21]. The force generated by the reabsorption of the solutes is what drives the reabsorption of water into the interstitium [22]. Urine’s ability to concentrate resides in the loop of Henle, which creates a hypertonic medullary interstitium [19,23,24]. It is in the thick ascending loop of Henle (TAL) where active transport of solutes takes place through different co-transporting channels (Figure 1).

In this process, K^+^ is the limiting substrate for NKCC2 channels and, in turn, for total electrolyte reabsorption [19]. Due to the correct K^+^ concentration gradient between tubules and the interstitium, a positive electrostatic charge is present in the tubular lumen. This charge allows the paracellular reabsorption of Mg^2+^ and Ca^2+^ [19,25]. The distal convoluted tubule (DCT) mediates the correct regulation of electrolytes and their secretion [26]. The DCT plays an important role in K^+^ secretion [27] and the transcellular reabsorption of calcium, via the transient receptor potential cation channel subfamily V member 5 (TRPV5) [25], and magnesium, via the transient receptor potential cation channel subfamily M member 6 (TRPM6) [28]. 

The reabsorption of Mg^2+^ in this segment depends on the proper function of a sodium-chloride cotransporter, the NCC channel (encoded by the *SLC12A3* gene), which is exclusively located in the cells of the DCT (see Figure 1) [26,29]. NCC, together with uromodulin and calcineurin, have been described as important regulators of magnesium homeostasis in the DCT [28]. In “salt-losing tubulopathies”, such as BS and GS, any alteration in this transport mechanism causes a loss of electrolytes that interrupts the global reabsorption process, resulting in a wide range of progressive alterations in renal physiology [20,30,31].

## 3. Molecular Basis and Clinical Features of the Diseases

Although GS and the different types of BS share most of their clinical symptoms, they have different etiologies. Their associated proteins reside in different parts along the nephron resulting in different pathophysiological mechanisms and compensatory phenomena regarding their loss of function. For example, poor TAL functionality associated with BS causes a decrease in electrolyte bioavailability, which leads to activation of the juxtaglomerular apparatus with secondary hyperaldosteronism and glomerular hyperfiltration due to afferent arteriole dilation [3]. 

This compensatory effect results in the hypertrophy of the juxtaglomerular apparatus and a constant elevation of renin and aldosterone in plasma [20]. In contrast, the typical salt loss of the DCT associated with GS causes volume contraction, leading to hyperaldosteronism with elevated plasma renin levels [25]. In GS, there is no direct involvement of the juxtaglomerular apparatus [20], which marks a relevant molecular difference from BS.

### 3.1. Molecular Basis of Gitelman Syndrome and Clinical Consequences

As mentioned previously, GS and BS share pathological conditions, including hypokalemia, hypochloremic metabolic alkalosis, hyperreninemia and secondary hyperaldosteronism [3,4]. However, the main difference is that GS also presents with hypocalciuria. This is due to an increase in Ca^2+^ reabsorption in an attempt to compensate for the loss of salts [25]. This compensatory process does not occur in patients with BS, since electrolyte dysregulation in the TAL causes a lack of the positive light gradient necessary for paracellular Ca^2+^ reabsorption [25,32]. Active transport of Ca^2+^ remains unaltered in GS [25,32].

Another relevant molecular characteristic is the presence of hypomagnesemia, mainly in cases of GS. However, this anomaly is not a complete differentiation between GS and BS [33] since it can also be found in certain cases of BS [34]. Although this phenomenon has been studied for more than 20 years [32,35,36], the reason for this magnesium loss is not completely understood. One of the most accepted hypotheses points to the alteration in the expression of the TRPM6 channel [28], the principal channel by which Mg^2+^ is reabsorbed in DCT (Figure 1). Moreover, there is a strict correlation between hypomagnesemia and chondrocalcinosis [37]. 

Mg^2+^ is a cofactor of the group of pyrophosphatases, particularly for alkaline phosphatase [38]. A decrease in the concentration of magnesium causes a dysfunction of these proteins, which increases the levels of pyrophosphate. Inorganic pyrophosphate binds to Ca^2+^ ions by ionic interaction, resulting in crystal formation. These crystals are deposited over time, eventually causing chondrocalcinosis [39]. In fact, magnesium is a crucial factor in the prognosis of GS, due to the possible development of such pathological conditions [37].

Historically, GS has been considered a benign disease, which was generally diagnosed incidentally by the presence of cramps, extreme fatigue, tetany or muscle weakness [5]. However, there is a wide phenotypic variability among patients, ranging from asymptomatic individuals to individuals with a severe phenotype [33]. Furthermore, it was considered as a “desirable disease” in the sense that patients typically have normal or low blood pressure [33,40]. For instance, a recent case report described a kidney transplant from a donor with GS, in which the possibility of lowering the blood pressure in the recipient was considered beneficial [41]. However, clinical surgeries like this can be seen as controversial. 

Similarly, GS (as well as BS) was considered as a human model of hypotension, since it was seen that carriers of pathogenic heterozygous mutations in their respective genes were associated with a lower blood pressure than that of the control population [42]. Interestingly, a study performed in 2000 had already anticipated that some pathogenic mutations in *SLC12A3* could protect against hypertension, which coincides with the literature [41,42,43,44,45,46]. Nevertheless, the presence of some homozygous variants is associated with primary hypertension [47]. A relevant study confirmed the possibility that Gitelman’s patients may develop hypertension due to continuous activation of the RAAS axis [48]. The role of zygosity in this process has yet to be elucidated.

In addition to hypertension, several phenotypes have also been related to this disease. For example, it has been stated that patients with GS have a greater predisposition to viral infections [49,50], and they are more prone to develop type II diabetes mellitus [28,51,52]. Moreover, it was recently postulated that they also have an abnormal glycosylation pattern in angiotensin converting enzyme 2 (ACE2), which leads to the activation of RAAS (also seen in BS patients) [45].

Long-term population studies might provide a better understanding and anticipation of the comorbidities that Gitelman’s patients might face. It is essential to make a great effort in the study of genotype-phenotype correlations, emphasizing the correlations with different levels of blood pressure values.

### 3.2. Molecular Basis of Bartter Syndrome and Clinical Consequences

In terms of presentation, Bartter Syndrome has been traditionally grouped into neonatal or classic Bartter Syndrome. The neonatal refers to a severe form with an antenatal presentation that leads to serious polyuria. Consequently, polyhydramnios, premature delivery and severe cases of electrolyte and water loss occur [53]. The classic type refers to a more subtle presentation that can occur at any time, but typically in early childhood, with polyuria, polydipsia, volume contraction and muscle weakness [34]. Currently, the different forms of Bartter Syndrome are classified into six subtypes based on the underlying gene. Moreover, the different subtypes can be further grouped into three categories, based on the similarity between the main molecular mechanisms in which the encoded products participate and their associated pathophysiology: BS type 1 and 2; BS types 3 and 4; and BS type 5.

Bartter syndrome types 1 and 2

The proteins NKCC2 (BS type 1) and ROMK (BS type 2) are the main players in the reabsorption of solutes in the TAL [19]. When they are disrupted, the physiological abnormalities lead to an early phenotype, usually with manifestations appearing during the prenatal stage [54]. The main consequences of its dysfunction in the embryonic stage includes electrolyte imbalances that can cause polyuria due to isosthenuria, polyhydramnios, preterm birth [55] as well as subsequent growth retardation, serious episodes of salt loss, hypercalciuria and metabolic alkalosis with hypokalemia and hypochloremia [54]. Although type 1 and type 2 share most of these symptoms, the appearance of episodes of early transient hyperkalemia is seen mostly in type 2 [56]. 

Newborns with mutations in the *KCNJ1* gene may not be able to excrete potassium via ROMK and other non-canonical mechanisms restore potassium levels later [54]. Due to the strict correlation and functional of the transporters in the TAL, the loss of function of the ROMK channels (BS type 2) could lead to the inactivation of NKCC2 (BS type 1) [57], which could justify the overlapping phenotype in both types. Furthermore, an interaction of both channels has also been demonstrated in the secretion of uromodulin protein [58,59], which is decreased in patients with type 2 Bartter [59]. Thus, these results point to a possible role for uromodulin in tubular disorders.

In addition, new studies on the function of NKCC2 channels have shown that stoichiometry can change due to mutations in the *SLC12A1* gene [60]. The type of mutation can determine differences in electrolyte disorders, which would explain the high phenotypic variability between individuals. For example, particular genetic alterations could mean that the Na^+^ K^+^ 2Cl^−^ cotransporter changes to the unique Na^+^ Cl^−^ transport. Genotype-phenotype correlation studies of large cohorts could help determine the relevance of the type of mutations in relation to the phenotype.

Bartter syndrome types 3 and 4

BS type 3 is caused by genetic abnormalities in *CLCNKB* gene. The transporter encoded by this gene mediates the reabsorption of chloride from tubular cells to peritubular capillaries in TAL and DCT [2,34]. This type of BS is one of the most investigated, due to the clinical similarity with GS and the need to differentiate them for an accurate diagnosis. It is characterized by enormous clinical variability. Thus, the first manifestations can appear at any time, from the antenatal to the adult stage, and truncating variants are mainly associated with early onset of the disease [34]. 

The central feature of BS type 3 is severe hypochloremia [54]. CLCNKB is one of the channels necessary for the reabsorption of NaCl [34] in TAL, and its function has to be intact for the reabsorption of chlorine in DCT [61,62]. In fact, its total genetic inactivation is incompatible with life [63]. An incorrect reabsorption can affect the functionality in the HCO3^−^/Cl^−^ exchanger [54], and thus chloride homeostasis is totally damaged in this segment of the nephron. CLCNKA cannot compensate for the loss of function in CLCNKB, on the basis that *CLCNKA* expression is decreased in orthologous *CLCNKB* null mice [61].

BS type 4 can be caused by mutations in *BSND* (type 4a) or digenic recessive mutations in CLCK– channels (*CLCNKB* and *CLCNKA*, Type 4b). In both cases, congenital deafness is a differential symptom [64,65], due to the loss of potential load in the inner ear and the incorrect function of chloride channels.

Furthermore, the BSND is not properly y localized when CLCNK channels are absent [63]. BSND is a mandatory subunit for the normal function of all CLCNK channels [12,65], so the phenotype of BS type 4 is more severe than BS type 3. Interestingly, although BSND is also present in DCT, BS type 4 is generally not confused with GS, as is the case with type 3 BS. In contrast to the other types of BS, renal failure is mainly associated with patients with type 4 BS [12].

Since the first discovery in 2004 of recessive digenic inheritance of the *CLCNKB* and *CLCNKA* genes [14,15], only a few cases of BS type 4b have been reported since then. This is because these types of cases, in addition to being rare, are not usually included in population-based studies of BS. This complexity can add another intriguing aspect. Given that large heterozygous deletions of the physically contiguous genes, *CLCNKA* to *CLCNKB* have been reported, the question remains, is CLCNKA actually associated with the disease? Until now, no homozygous or heterozygous patients with *CLCNKA* have been phenotypically reported. 

The homozygous inactivation of the orthologous *CLCNKA* gene in animal models resembles the phenotype of diabetes insipidus, although the mice do not lose salt [66]. As in the case of Alport syndrome associated with the contiguous deletion of the *COL4A5* and *COL4A6* genes [67], excluding the association of the *COL4A6* gene with said syn-drome [68], it would be interesting to identify the clinical or molecular relevance of the inactivation of the *CLCNKA* gene alone in BS. Recently, the p.R83G variant in *CLCNKA* has been postulated as the putative gene loci for a major incidence of heart failure in dilated cardiopathy [69]. Except for this association, thus far, no possible phenotypes directly related to the functional deficiency of the *CLCNKA* gene have been identified.Bartter syndrome type 5.

The clinical characteristics of patients with mutations in the *MAGED2* gene are very similar to classic Bartter, highlighting polyuria, hyperreninism and hyperaldosteronism. However, the most relevant clinical findings consist of severe polyhydramnios, premature birth and perinatal complications. Despite starting as a severe form of Bartter syndrome [13], phenotypic restoration occurs spontaneously, without the need for any specific treatment. 

The explanation for this temporal manifestation lies in the fact that the apical localization of NCC (*SLC12A3*) and NKCC2 (*SLC12A1*) depends on MAGED2 during the developmental stages in humans [13]. It is logical that the diminished ubieties for both cotransporters bring about a severe phenotype [70]. Two theories have been put forward on the pathophysiology related to MAGED deficiency. The first is that MAGED2 binds to Hsp40 to regulate endoplasmic reticulum-associated degradation (ERAD), and therefore mutations in *MAGED2* cause alterations in this process [71] and NCC and NKCC2 are retained intracellularly. 

The second postulates that the mutations in *MAGED2* prevent sufficient concentrations of cyclic adenosine monophosphate (cAMP) for the correct function of the antidiuretic hormone (ADH), causing the mislocalization of the channels [70]. It is essential to bear in mind that ERAD, lysosomal degradation and the specific ubiquitination of unfolded and immature NCC and NKCC2 have also been widely described as key mechanisms of protein expression and localization [72,73,74,75], for which further study of Bartter’s disease type 5 will contribute to a greater understanding of these fundamental processes. Similarly, the presence of mutations in the *MAGED2* gene could explain the cases of idiopathic polyhydramnios and unresolved prenatal tubulopathies [76]. 

### 3.3. Long-Term Outcomes in GS and BS

Investigation of tubular channels as well as micropuncture and patch clamp studies have significantly benefited our understanding of GS and BS [27,77,78,79,80]. Despite this, it is still necessary to continue contributing to basic research on these pathologies. The different studies that focused on the characterization of these diseases in various stages of the disease had small sample sizes. Therefore, many unknowns about the progression of these diseases and their possible complications remain to be clarified. 

For example, it is necessary to investigate the systemic consequences of these tubulopathies. Recent studies suggested that *SLC12A1* (NKCC2) could be involved in the pathogenesis of hyperparathyroidism [81,82]. The current evidence is not sufficient to establish the link, since the sample sizes only reached a dozen patients, and the biochemical relationship is unproven. It would be interesting to expand these clinical case studies, paying particular attention to the interactions between NKCC2 and CASR (extracellular calcium-sensing receptor) (the molecular target of drugs for hyperparathyroidism) [82]. Recent studies have suggested that GS [83,84,85] and BS [34,85,86] may be associated with glomerular damage, despite being classic tubulopathies. In both cases, moderate proteinuria associated with focal lesions of segmental glomerular sclerosis and thickening of the glomerular basement membrane have been reported [84]. These alterations could be due to possible glomerular damage due to sustained tubular insufficiency, as occurs, for example, in Dent’s disease [87]. This can lead to an erroneous clinical diagnosis, and for this reason it is very important to consider certain diagnostic tools for a correct clinical diagnosis.

## 4. Diagnostic Approaches

Figure 2 summarizes the main diagnostic methods for Gitelman and Bartter syndromes, as well as the relevant aspects of each to take into consideration.

### 4.1. Clinical Diagnosis

A differential clinical diagnosis based solely on symptomatology is currently considered difficult and inaccurate. Since there is phenotypic overlap between GS, and the different types of BS, as well as great clinical variability between individuals with the same syndrome. Generally, a prenatal presentation of BS can automatically rule out BS type 3 however, BS type 3 can, in some cases, lead to an early or prenatal manifestation of the disorder [54]. 

Adding a further degree of complexity, there are other clinical entities that resemble BS and GS that can be confused and thus lead to inaccurate diagnoses. Among others, autosomal dominant familial hypocalcemia stands out, which before the discovery of *MAGED2,* it was considered to be the fifth type of Bartter syndrome [25]. Familial hypocalcemia is caused by pathogenic mutations in the gene *CASR*, these include activating mutations (OMIM #601199). This gene encodes a plasma membrane G protein-coupled receptor, CASR, which in the kidney tubule negatively regulates calcium resorption. The inappropriate activation of the CASR receptor by these activating mutations leads to hypochloremic metabolic alkalosis, hypokalemia and hypocalcemia, a phenotype that mimics BS [88]. 

Moreover, the inadequate activation of this receptor entails a reduction in the activity of other channels involved in BS, including ROMK [88], NKCC2 [89] and Na^+^/K^+^ ATPase [90,91] therefore, this resembles the loss of function of NKCC2 and NCC accomplished by mutations in *MAGED2* (previously discussed) [88]. Of note, the dominant inheritance pattern of familial hypocalcemia differs from the characteristic pattern of BS which is recessive [92]. 

Given the degree of complexity associated with the physiology of these syndromes, it is important to prioritize clinical peculiarities that, in some cases, allow a differential diagnosis, as well as to identify symptoms that characterize other conditions similar to GS and BS that may be confused with each other [93]. Some of the most relevant indistinguishable conditions are summarized in Table 2 (for GS) and in Table 3 (for BS). Furthermore, in Figure 3 we summarize the best differential symptoms that could help to discern between GS and the different types of BS. 

Generally, a pattern of electrolyte alterations in the biochemical studies support the first proof for the clinical diagnosis [94]. Additionally, an elevation in blood of renin, aldosterone and prostaglandin E_2_ (PGE_2_) are specific parameters that can shed light on the precise clinical diagnosis for GS and BS, disregarding other tubulopathies [95]. One possible parameter that can provide a differential diagnosis of Bartter and Gitelman over other diseases that exhibit metabolic alkalosis, includes the assessment on Cl^−^ concentration in urine, which is persistently high [96]. Moreover, chloride status is considered as an important biomarker before the start of any pharmacological treatment [85].

### 4.2. Diuretics Tests

Diuretics are active natriuretic principles that modify electrolyte transport and are widely used for the treatment of hypertension, heart failure, and fluid overload [120,121]. The main idea of conducting a diuretic test is to demonstrate the lack of effect in patients with GS and BS [122,123]. Specifically, the lack of effect is observed for the diuretics that block the channels related to the diseases; thiazide diuretics for GS and loop diuretics for BS (Figure 1). Briefly, thiazide diuretics cannot manifest its effects in GS because NCC channel has an altered function. In BS, loop diuretics (furosemide as a classic example) will have no effect because the NKCC2 channel is not at its optimal activity. To analyze this effect, the natriuretic and chloruretic effect of the drugs are assessed in spot urine samples, approximately 6 hours after administration [122,123]. The main objective of the clinical trial NCT00822107 was to evaluate the chloruretic response to thiazide in Gitelman syndrome. The results were published in April 2019 and revealed that the thiazide response in GS was null, but only five patients were analyzed (all of them women). This should be considered a clue, but it does not reflect the usefulness of taking these types of exams. Likewise, clinical information rescued by these tests is not totally enlightening. 

On the one hand, it is difficult to obtain an absolute answer (yes or no) to a drug treatment. This will only be useful for GS (*SLC12A3*) or BS type 1 patients (*SLC12A1*), being the remaining types of BS only partially interpreted. On the other hand, it is controversial to withdraw daily treatments to patients (K^+^ sparing diuretics) to perform the diagnostic test [124]. In addition, other treatments cannot be withdrawn (such as antiprostaglandins or antialdosterone) and this could introduce a relevant bias [123]. Finally, in the case of BS, diuretic tests also carry possible serious side effects, such as severe volume depletion [125]. For GS, this consequence is not very serious but there is a wide variability in the response for diuretic drugs in GS [123], that can lead to misinterpretation.

### 4.3. Analysis of Urinary Microvesicles

Another possible recent clinical approach has been the study of urinary microvesicles in order to diagnose and predict possible biomarkers of GS and BS [126,127]. All the channels mainly related to these diseases are apical proteins that can be manifestly secreted into multivesicular bodies (MVBs). Therefore, the vesicular proteomic profile of each patient can be studied using different proteomic techniques, in order to identify possible biomarkers of disease. Despite being a non-invasive method, this approach is not yet a reality in current clinical routine due to its cost and time [128].

### 4.4. Genetic Diagnosis

Given the high phenotypic inter- and intra-familial variability of these syndromes and the large number of phenotypes that often lead to confusing them, genetic studies for GS and BS may be the most conclusive tool for diagnosis. The identification of causal mutations in any of the associated genes can not only determine the diagnosis, but also allow for establishing a better prognosis of the disease [37,54] and a better treatment strategy for the patient [33,125]. 

In addition, information related to mutations could help to inform about possible comorbidities. From a public health point of view, genetic studies are also the basis for prenatal and preimplantation diagnosis. Prenatal tests in BS are very relevant in prenatal forms of types 1, 2, 4 and 5. In fact, any clinical case with polyhydramnios without morphological anomalies must raise suspicions of Bartter syndrome, and the case is candidate for genetic testing [13,34,71]. 

Additionally, in GS, prenatal tests would be of great value [129] since some cases have been reported and diagnosed during pregnancy [130,131], with episodes of hypokalemia and salt craving. Nevertheless, as we will discuss later, genetic testing is not without problems and a conclusive diagnosis is not always achieved.

The development and constant improvement of different techniques for the study of the genome, such as massive sequencing technologies (ultra-sequencing or Next Generation Sequencing-NGS) and the continuous reduction in cost allows more health systems to include this type of test as part of the diagnostic routine. The process of searching for pathogenic mutations generally begins with the sequencing of the known associated genes by NGS technologies. The GS and BS genes are often part of predesigned panels of known genes involved in tubulopathies [2], chronic hypokalemia or salt-acid disorders [5].

Another complementary method is the search for CNVs (copy number variations) in candidate genes, through specific NGS products and software, arrays of SNPs (single nucleotide polymorphism) or MLPA^®^ (Multiplex Ligation-dependent Probe Amplification), among others [12,132,133]. This method is of particular interest for *CLCNKB* gene, where a large deletion was found to be the most common mutation, particularly among the Chinese population [34,54,132,134,135]. This strategy will normally end up with the identification of candidate diagnostic mutations in one of the BS or GS associated genes. 

A recent study demonstrated an efficacy greater than 85% in finding the causal BS mutations [76]. For GS, the diagnostic efficacy is lower, around 62% [5,136,137,138,139], and a considerable number of clinical-diagnosed patients carries only one mutation (around 25%) [136,137,138,139]. In both scenarios, it is recommended to carry out a study of carriers of the variants identified in the parents, in order to determine that each variant is located in different copies of the gene. In a subset of the cases studied, an inconclusive or negative result can be obtained. 

An inconclusive result refers to studies in which only one candidate variant is found, or where the variant or variants found are classified as “variants of uncertain significance” (VUS). VUS are variants for which there is not enough knowledge to predict whether they could cause a pathogenic effect on the organism, and thus it is not possible to unequivocally establish a causal relationship between the variant and the patient´s disease. Therefore, they cannot be classified as pathogenic or neutral [140].

The lack of identification of a second mutation or the absence of any variant may be due to different situations: (a) the nature of the mutation it is not detected by the technology used; (b) the pathogenic mutations are in regions of the genome, with an unknown, or little-known function. These regions are generally excluded from analysis due to the difficulty of predicting their impact on the phenotype [141]. This may be the case of variants that affect promoters, poly (A) regions, non-translated regions (UTR) or deep intronic regions, among others; (c) the causative gene has not yet been discovered and is therefore not included in diagnostic gene panels. Investigation of unresolved cases and their parents using whole genome strategies could lead to the discovery of new genes associated with GS or BS; or (d) the identification of a second mutation is not described since it does not really exist. 

As with other disorders, it could be the case that the presence of a single heterozygous pathogenic variant located in a gene associated with recessive inheritance is associated with mild forms of the disease. Advances in basic research are providing new insights into the function of hitherto poorly understood genomic region, which may lead to improved interpretation of the effect of variants in these regions [142]. In the same way, the constant improvement in the technology for studying the genome will probably increase the rate of positive studies.

To date, no large populations of GS have been carried out that allow establishing a representative genotype-phenotype correlation of the disease [50]. However, peculiarities of different populations have been identified, such as, such as hotspot mutations in the Japanese population associated with higher levels of magnesium in the blood than others [50]. Another example is the exclusivity of two mutations in the Roma ethnic group: *SLC12A3* c.1180+1G>T and *SLC12A3* c.1939G>A [143]. 

For BS there is a defined phenotype for each subtype; however, these manifestations may overlap with others, especially early in the disease [54]. In any case, there is not a strict correlation for the type of mutation and the disease severity, as in the case of other inherited diseases such as Autosomal Dominant Polycystic Kidney Disease [144,145]. Perhaps, this is due to the lower prevalence of GS and BS, which makes it more challenging to study.

#### Heterozygous Carriers for Gitelman Mutations

Individuals carrying only one heterozygous pathogenic mutation are not considered as patients of this disease. However, as with other pathologies, it has been described that these patients may present a mild phenotype [51,146,147,148,149]. It is arguable that these individuals might have a second mutation, but several independent studies have ruled this out. In a study carried out in 147 carriers in the Amish population [146], lower blood potassium values were associated with heterozygous carriers. 

However, a recent cross-sectional study (with 81 heterozygous individuals) did not reflect this difference. The lower K^+^ value shown in this article [146] could be related to the type of mutation (*SLC12A3* p.R642G, rs200697179). However, correla-tion of genotype-phenotype has not been established in GS to date, and only recent studies are beginning to determine these variabilities [50]. Another possibility is that either genetic background in this population or environmental-acquired factor might also play a key role. 

As in previous studies, recent publications [51,146,147] highlighted higher resting heart rates as the main characteristic of heterozygous carriers. In contrast to previous studies, they did not identify lower values in blood pressure for heterozygotes [148,149]. In one of them, the lower blood pressure was only referenced in children [149]. Although there is a current debate on this topic, the general conclusions thus far do not relate lower blood pressure in heterozygotes. Therefore, clinical peculiarities ascribed to *SLC12A3* in heterozygotes may not result in a pathological condition but will be paired with a different clinical profile. In any case, this situation supports a genetic complexity in GS, rather than it being a classical recessive inherited disease.

## 5. Therapeutic Approaches

### 5.1. Current Pharmacological Treatments

GS and BS are determined by the lack of clinical interventions and therapeutic possibilities. There is not a definitive treatment and therapeutic interventions are based on supportive treatments. Furthermore, information on the efficacy of each therapy is scarce that there is not a well-established algorithm for the treatment of these patients [125]. Thus, each patient receives a treatment based on the individuals’ effectiveness [20]. The first goal of any treatment for these syndromes is the restoration of the electrolyte profile. Due to the genetic nature of the diseases, these disorders are chronic, and the treatment is for the entire life of the patient. 

#### 5.1.1. Oral Salt Supplementation

Water and salt supplements are the first-line approach since their goal is to restore electrolyte loss. Although the commercially available supplements have improved since ab libitum salt treatment [33], patients face several problems with such formulations. For example, the number of tablets and pills per day is often high, which creates an uncomfortable daily situation. An examination of the electrolyte abnormality profile of each patient is necessary to prescribe adequate supplementation. KCl is the preferred form of salt supplementation because it replaces the loss of potassium and chloride (both of them are affected) [85]. 

In patients with hypomagnesemia, magnesium supplementation becomes very important to prevent associated chondrocalcinosis [150]. Magnesium values are not easily restored and the bioavailability is quite variable depending of the quelant for Mg^2+^, causing adverse effects and ongoing inconveniences such as diarrhoea [151]. Within salt supplementation, it is cruicial to assist the diet that patients follow carefully. The reason is that patients often have an appetite for salty foods, and continued and uncontrolled intake of salts can worsen the disease state [95]. In addition, traditional herbal medical preparations and dietary supplements can be enriched in salts and must therefore be administered with care or prohibited [24]. Pharmacokinetic studies would be very useful for these patients to establish the bioavailability achieved by each supplement in each individual. 

#### 5.1.2. Non-Steroidal Anti-Inflammatory Drugs (NSAIDs)

Non-steroidal anti-inflammatory drugs (NSAIDs) are medicines widely used to relieve pain, reduce inflammation and lower the temperature [152]. Its mechanism of action is based on the non-selective inhibition of the cyclooxygenase (COX), the enzyme that is involved in the synthesis of PGE_2_ [153]. Therefore, they are useful to prevent the elevation of PGE_2_ that typically occurs in BS patients, as the ones that benefit the most from this treatment compared with GS patients. However, these drugs are used in both diseases, as they can alleviate hyperprostaglandinuria, secondary hyperaldosteronism, hypochloremic hypokalemic metabolic alkalosis, polyuria, hypercalciuria and growth retardation [154]. 

Indomethacin is the most prescribed NSAID for the treatment of BS [125]. Its use has showed strong benefits and is crucial at an early start in the treatment. The most relevant effect is the restoration of electrolyte abnormalities, which improve the development of the disease. Another important point is the reduction of nephrocalcinosis since indomethacin is able to reduce the hypercalciuria [155,156]. 

Although the symptomatology of GS usually appears during later in childhood or adulthood, there is wide variability in the spectrum of GS and symptoms can appear early in childhood [157]. In these cases it is common to suffer failure to thrive. Hence, the use of indomethacin could result as beneficial as for Bartter´s patients. This drug was also effective in the correction of hypokalemia during adulthood in a study that included 22 Gitelman´s patients [158]. Indomethacin showed better results than eplenerone or amiloride (discussed later). 

The paradox of these treatments is the unavoidable adverse effects of NSAIDs on the kidney [159,160] and gastrointestinal tract [161], even when administered with proton pump inhibitors, such as omeprazole [125,152]. Gasongo et al. [156] suggested the monitoring of renin concentration to achieve the minimal dose with which it is possible to see a beneficial effect. This proposal points towards a more personalized medicine for each patient and could be strongly advantageous. This proposal is particularly important for affected children with different needs for indomethacin or ibuprofen treatments to avoid developmental delays. In this scenario, hospital pharmacies are very useful to prepare the adequate oral suspensions, in relation to the weight and dose required by these patients [162].

The toxicity of NSAIDs is a logical reason to try alternative medications such as COXIBs (ciclooxygenase-2 inhibitors) which, in contrast to NSAIDs, can selectively inhibit COX-2 [163]. However, COXIBs can also cause adverse effects, such as an increased risk of cardiac problems and, in particular, myocardial infarction [164]. To date, there are no well-documented records of secondary effects under the treatment of COXIBs in BS or GS patients. Thus, there is a strong preference to NSAIDs over COXIBs [165], and COXIBs only supports a better therapy when the use of NSAIDs plus proton pumps inhibitors causes an uncontrolled severe hypomagnesemia [125]. 

In summary, NSAIDs constitute one of the most reliable treatments for the symp-toms of BS and GS, despite not being a definitive pharmacological approach.

#### 5.1.3. Potassium Sparing Diuretics

Potassium sparing diuretics include a series of diuretics without a strong diuretic effect. However, they are capable of restoring hypokalemia caused by diuretics furosemide and thiazide, this being the main reason for their use [166,167]. They act in the distal part of the nephron and are classically subdivided as aldosterone antagonists or Na^+^ channel blockers [166]. Considering the suppressive effect of these drugs on aldosterone and secondary hyperkalemia, they can be considered as promising pharmacologic agents in GS and BS. However, if the distal region is blocked, partial reabsorption of Na^+^ and water cannot occur, leading to worsened salt loss. Therefore, the patient could be at risk of life-threating hypovolemia [20,168,169,170]. 

The first investigations that deepened the usefulness of these drugs found good efficacy in the use of amiloride in patients with BS [171,172,173]. Notably, amiloride was reported during pregnancy in a few cases [174,175] (B category for pregnancy [176]). However, a recent study reiterated the dangerous consequences of amiloride treatment, including hypovolemia and fibrosis in the kidney and heart in long-term treatments [158]. Another complication of its discontinuation is hypotension [169,170]. On the contrary, a beneficial point of its use is a minor loss of magnesium with this treatment, although only slight modifications were observed [158]. 

Essentially, sparing diuretics are only accepted clinically when the other therapeutic options, such as NSAIDs or supplemental drugs are have no impact on phenotype improvement [170,171]. Moreover, cases with a severe phenotype do not improve even with these treatments [177] and the risk of hypovolemia must be carefully assessed [20,168,169,170].

#### 5.1.4. Renin-Angiotensin-Aldosterone System Inhibitors

GS and BS are characterized by a constant activation of the RAAS axis in an attempt to recover lost salt and water [25]. In fact, this alteration is the cause of both hyperreninism and hypertrophy of the juxtaglomerular apparatus (specific for BS) [20]. Hence, the use of inhibitors of this axis was proposed as a therapeutic strategy. This clinical approach is similar to sparing diuretics, because the potential risk of hypovolemia and hypotension caused by these medications causes restrictions in the prescription of these drugs [20,178]. ACE inhibitors cannot resolve the clinical abnormalities of GS and BS patients. In fact, they do not improve electrolyte disturbances [179]. Despite this, they can be a therapeutic possibility for cases of refractory NSAIDs, nephrotoxicity with NSAIDs [180] or even superior to sparing diuretics [181]. In addition, a possible benefit is the partial correction of hypomagnesemia, which was described in one case with enalapril [179].

#### 5.1.5. Growth Hormone (GH)

Growth hormone (GH) is intended to prevent growth abnormalities [182]. Im-portantly, GH is considered only after the failure of NSAID treatment [125] for the same purpose. GS is a disease in which growth retardation is less frequent than in BS [33] but growth hormone is considered as beneficial [33,50]. Perhaps, GH shows a more beneficial effect in GS than in BS due to the worse phenotype in BS in terms of growth retardation. In fact, Fujimura´s studies [50] stated that GH treatment was not accurate in all patients. The duration of treatment may need to be longer to achieve visible efficacy even into adolescence or young adulthood. 

This strongly correlates with clinical guidance, in which treatment should be prolonged until a coherent mineral density is achieved [183,184]. To date, GH has been proven in cases of kidney damage and does not impact further damage [185]. However, the major drawback is the controversy among endocrinologists over the side-effects of the long-term use of the drug. The most recent guidelines [186,187] have pointed out two essential points: The relevance of genetic testing to ensure the causes and the possibility of discontinuation and the improvements in the adherence with lower frequencies in the administration of the recombinant human growth hormone. 

#### 5.1.6. Overview of the Effectiveness of These Treatments

It is important to note that all the current treatments are supportive, and none of them can solve the main problems later in life. Therefore, more research essential to provide better solutions for the patients. Another key point is that BS and GS therapeutics overlap, sharing the vast majority of possible treatments as reflected in Table 4. 

Treatment in GS and BS is based on the administration of a wide range of drugs, which depends both on the criteria of the physician and on the severity of each patient [33,125]. Therefore, clinicians pay special attention to the manifestations of hypomagnesemia and its consequences in GS, while in the case of BS, indomethacin is the main approach [125]. In this scenario, there is no comprehensive understanding of the disease or greater improvements in the phenotype of the patients. In fact, knowledge is acquired through description of cases from independent reports, which is why large population studies are required. 

On the other hand, currently there is still debate on different topics. There is no consensus on the treatment of practically asymptomatic patients with GS [95] or the definition of the threshold that symptomatic patients with salt-wasting tubulopathy must reach to ensure a good prognosis [20]. For example, one question is what level should be reached to have a correct intracellular magnesium concentration in GS [188]. In this sense, the basic concepts in a treatment, such as the “treatment objective” are not yet defined. The "goal of treatment" is to define the point at which treatment can be interrupted or discontinued [189]. A relevant study of A. Blanchar et al. [158] argues that a later period of treatments may improve efficacy. 

Perhaps, the correction of the electrolytes in the blood is a gratifying achievement, as it is the most reliable surrogate endpoint, however, another specific point to consider should be the prevention of renal and extrarrenal manifestations such as end stage renal disease (ESRD), chondrocalcinosis or mineral bone fractures. In conclusion, therapeutic possibilities in GS and BS are dominated by a paradigm with fairly variable prognosis.

Currently, despite the lack of knowledge regarding prognosis, this is usually acceptable with correct adherence to the respective treatments [85,190]. Gitelman´s patients have an acceptable prognosis [85], although asthenia and muscle weakness are common daily manifestations which impair quality of life [33]. Bartter´s patients with early presentations can start treatment with indomethacin in the 4th–6th week of life, showing a good response and adequate prognosis [190]. However, BS type 4 shows the worst phenotype, being possible the appearance of nephrocalcinosis and progression to ESRD [191,192]. There is no information on the prognosis in the case of lack or suspension of treatments or the influence of a delayed diagnosis and treatment. Furthermore, there is no information on the differences among different therapeutic combinations in terms of prognosis.

### 5.2. Future Therapeutic Perspectives

The clinical trials for GS and BS do not offer new drugs for their treatment. In fact, the trials inlvolve different combinations of commonly used drugs, and there is no drug with a newly transitional mechanism of action [193,194]. 

One of these studies uses acetazolamide (a diuretic that inhibits carbonic anhydrase [195]) to alleviate the metabolic alkalosis of BS (NCT03847571). Although the results have not been yet published, a recent article from the same group suggests the use of acetazolamide as another possible adjuvant of the classical therapies, never as monotherapy [196]. 

Another study has the intervetion of progesterone to alleviate the hypokalemia (NCT02297048). In line with this hypothesis, Blanchard et al. [197] have reported that adrenal glands adapt their response to levels of hypokalemia and hyponatremia. This could be the first step towards a better understanding of the pathophysiology and, perhaps, towards a new therapeutic group.

## 6. Conclusions

GS and BS are inherited tubulopathies with challenges in their precise diagnosis since both pathologies have overlapping clinical symptoms. Regarding GS, it used to be considered as a mild disease, but patients frequently report extreme fatigue, showing an impaired quality of life. Regarding BS, these patients tend to have a worse clinical outcome. In fact, symptoms can appear before birth, resulting in delayed growth and propensity to ESRD.

In terms of diagnosis, genetic tests, regardless of the difficulties they may have, are the optimal tool for an accurate and definitive diagnosis. On the other hand, prognosis is poorly known currently, and therefore, population studies that investigate following up with patients are a huge necessity. Lastly, basic research enables the discovery of the molecular basis of the pathologies as well as the identification of novel therapeutic targets.

## Figures and Tables

**Figure 1 ijms-22-11414-f001:**
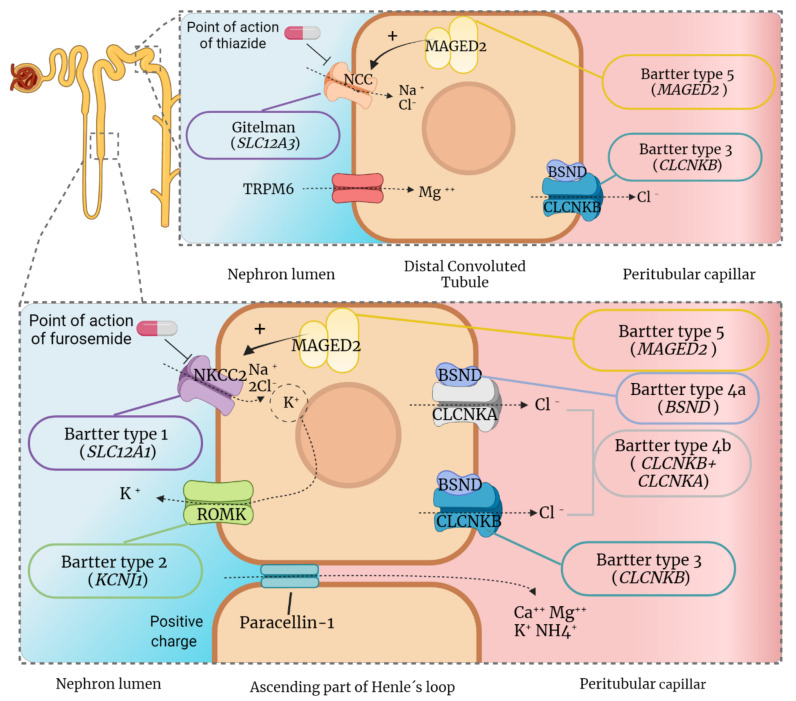
Proteins and channels implicated in the pathogenesis of Gitelman and Bartter syndromes. The electrolyte transports of the most important channels for the diseases are represented as well as the channels related to the inhibition by thiazide (NCC) and furosemide diuretics (NKCC2). Each disease is accompanied by the causative gene (in capital letters, brackets and italics), whereas the corresponding protein is indicated above the channel (only in capital letters). NCC: Solute carrier family 12 member 3; MAGED2: Melanoma-associated antigen D2; TRPM6: Transient receptor potential cation channel subfamily M member 6; CLCNKB: Chloride channel protein ClC-Kb; NKCC2: Solute carrier family 12 member 1; BSND: Barttin; CLCNKA: Chloride channel protein ClC-Ka; and ROMK: ATP-sensitive inward rectifier potassium channel 1. The positive charge of the DCT makes an electrochemical gradient from the luminal tubule from the interstitium possible.

**Figure 2 ijms-22-11414-f002:**
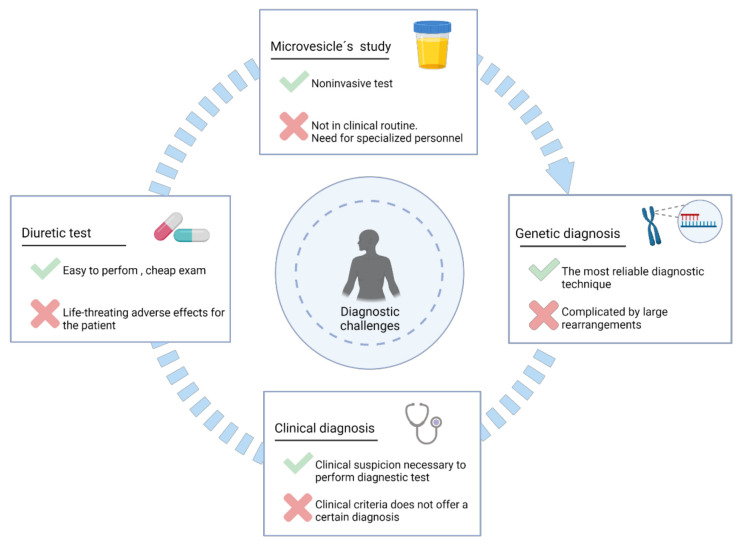
Main methods for the diagnosis of Gitelman and Bartter syndromes. Each method is accompanied by its principal benefit and the major drawback.

**Figure 3 ijms-22-11414-f003:**
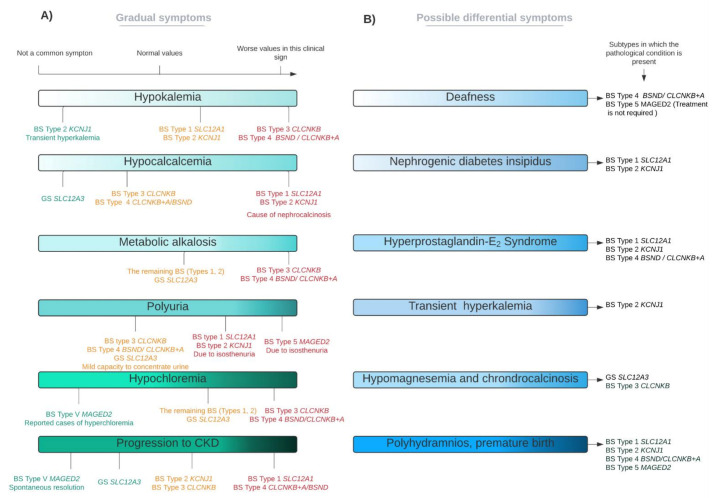
The phenotypic spectrum of GS and BS; principal signs. This figure reflects the high variability that exists among the different subtypes of BS and GS. (**A**) Clinical parameters present in several subtypes, with different levels of severity. Red colour is indicative of a worse severity, whilst orange and green correspond to a lesser disease severity (in this descending order). (**B**) Characteristic (but not necessary) clinical parameters of specific subtypes, which can be used for differential diagnosis (if present). For these cases, there are not differential grades in each subtype.

**Table 1 ijms-22-11414-t001:** Bartter syndrome classification, associated genes and encoded proteins. All the UniProtein codes are referred to Homo sapiens. AR: Autosomal Recessive; DR: Digenic Recessive; and XLR: X-Linked Recessive.

Bartter Subtype	OMIM	Inheritance	Causative Gene	Related Protein	UniProt Code
Type 1	601,678	AR	*SLC12A1*	NKCC2 (Solute carrier family 12 member 1)	Q13621
Type 2	600,359	AR	*KCNJ1*	ROMK (ATP-sensitive inward rectifier potassium channel 1)	P48048
Type 3	607,364	AR	*CLCNKB*	CLCNKB (Chloride channel protein ClC-Kb)	P51801
Type 4a	602,522	AR	*BSND*	BSND (Barttin)	Q8WZ55
Type 4b	613,090	AR DR	*CLCNKB +* *CLCNKA*	CLCNKB (Chloride channel protein ClC-Kb) + CLCNKA (Chloride channel protein ClC-Ka)	P51801 + P51800
Type 5	300,470	XLR	*MAGED2*	MAGED2 (Melanoma-associated antigen D2)	Q9UNF1

**Table 2 ijms-22-11414-t002:** Clinical entities that mimic Gitelman syndrome.

Gitelman-Like Diseases	Clinical Similarities to Gitelman	Clinical Differences to Gitelman	References
Abuse of thiazide diuretics	Hypochloremic metabolic alkalosis Hyperaldostheronism Hypokalemia Hypomagnesemia	The symptoms will be ruled out after the withdrawal. Hyperchloruric response as indicative parameter	[3,96,97]
ADTKD-HNF1-β	Hypomagnesemia Hypokalaemia Hyperparathiroidism Young adult presentation (but not mandatory at this ages)	Dominant inheritance pattern Extrarenal manifestations: MODY type 3 Renal cysts	[98,99,100]
Cystic fibrosis	Hypochloremic metabolic alkalosis Hypokalemia Hypochloremia	Levels of chloride in response to treatment Similarities are present with hot weather	[101,102]
Autoinmune diseases; SLE ^1^, Sjögren syndrome, autoimmune thyroiditis	Muscular weakness Hypokalemia Hypomagenesemia Hypocalciuria Cramping	Presence of auto-antibodies against NCC ^2^	[103,104]
Congenital Chloride Diarrhea	Hyponatremia Hypochloremia	Low chloride in urine but high in stool Watery stool, similar to urine	[105,106,107]

^1^ SLE: Systemic Lupus Erythematosus; ^2^ NCC: Solute carrier family 12 member 3.

**Table 3 ijms-22-11414-t003:** Clinical entities that mimic Bartter syndrome.

Bartter-Like Diseases	Clinical Similarities to Bartter	Clinical Differences to Bartter	References
Abuse of loop diuretics	Hypochloremic metabolic alkalosis Hyperaldostheronism Hypokalemia Hypercalciuria Ototoxicity Hyperuricemiaç	The symptoms will be ruled out after the withdrawal. Hyperchloruric response as indicative parameter	[3,96,97]
Autosomal dominant hypocalcemia due to *CASR*	Hypokalemia Metabolic alkalosis Hyperreninemia, hyperaldosteronism Hypocalcemia with hypercalciuria Nephrocalcinosis	Dominant inheritance pattern Low concentration of PTH ^3^	[88,108,109]
ADTKD-*REN*	Early presentation in life (childhood) Hyperuricemia Hypokalemia	Dominant inheritance pattern Hypoaldosteronism Hyporeninism	[110,111]
Nephrogenic diabetes insipidus	Polyuria Failure to thrive Hypokalemia, hypercalciuria Nephrocalcinosis	Hypernatremia, hyperchloremia No response to desmopressin	[112,113,114]
Cystic fibrosis	Hypochloremic metabolic alkalosis Failure to thrive Hypokalemia Hypochloremia	Levels of chloride in response to treatment Similarities are present in hot weather	[101,102,115]
Dent’s disease	Hypokalemic metabolic alkalosis Hypercalciuria Hyperreninism Hyperaldostheronism	Proximal characteristic Proteinuria with low molecular weights	[87,116]
Congenital Chloride Diarrhea	Polyhydramnios Premature delivery Hypokalemia Loss of Na^+^ and Cl^−^ in urine	Low chloride in urine but high in stool Watery stool, similar to urine	[105,106,107]
Aminoglycosides antibiotics: gentamicin and amikacin	Metabolic alkalosis Hypocalcemia Hypomagensemia Polyuria Hearing loss (amikacin)	Slow recovery after the finish of the treatment	[96,117,118,119]

^3^ PTH: Parathyroid hormone.

**Table 4 ijms-22-11414-t004:** Currently used pharmacological treatments for Bartter and Gitelman syndromes.

Therapeutic Approaches	Gitelman Syndrome	Bartter Syndrome
Supplemental electrolyte drugs	Mandatory, especially with magnesium loss	Mandatory
NSAIDs	Possible	Indomethacin as principal treatment in BS
Potassium Sparing Diuretics	Possible, but not recommendable	Possible, but not recommendable
Inhibitors of RAAS axis	Poorly described, but possible	Possible, especially with nephrotic damage from NSAIDs
Growth Hormone	Possible, poor evidence of efficacy	Possible, poor evidence of efficacy

## Data Availability

Not applicable.

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
