# Peer review of "Molecular Basis, Diagnostic Challenges and Therapeutic Approaches of Bartter and Gitelman Syndromes: A Primer for Clinicians"

_ijms, 2021, doi:10.3390/ijms222111414_

Round 1

Reviewer 1 Report

Dear Authors, I enjoyed reading your manuscript, it is well-researched and comprehensive. However, please see my suggestions below to improve it:

  • please correct grammar, punctuation and formatting, as they are many errors in the manuscript (!!!)
  • re-read the manuscript and correct all the typos (they are numerous)
  • please standardize the way you present abbreviations - abbreviation first and explanation in parentheses or full name first + abbreviation in parentheses; also, I believe that a list of abbreviations would come handy to the reader
  • instead of Mg ++ or Ca ++, I think it should be Ca2+ (with superscript); also, please pay attention to the use of subscrpit and superscript throughout the manuscript
  • the figures would benefit from captions explaning the use of italic  and even the explanation of some of the abbreviations used.
  • please separate table ticle from its footnotes which should contain the abbreviations used and other explanations
  • I believe that you should re-write the Conclusions section to better underline the point of the paper and highlight its strenghts.

Author Response

We are pleased to see that the reviewer was generally enthusiastic about our article, though he/she has requested to add more information about some aspects. We agree that including this information would strengthen the manuscript and thus we have performed all the requested suggestions and corrections. We also wish to thank the reviewer for his/her helpful comments and suggestions to improve our manuscript.

Please find our “point-by-point” responses to the specific questions raised by the reviewer#1.

  • Please correct grammar, punctuation and formatting, as they are many errors in the manuscript (!!!)

We completely agree with reviewer#1 and we have corrected all the punctuation and formatting errors in the text. Also, all the text was completely proofread by a native English speaker to improve the text. We hope that the article is now in the correct format.

  • Re-read the manuscript and correct all the typos (they are numerous)

In line with the aforementioned comment and this one, we totally agree with reviewer#1 and we have changed all the previous typo errors.

  • please standardize the way you present abbreviations - abbreviation first and explanation in parentheses or full name first + abbreviation in parentheses; also, I believe that a list of abbreviations would come handy to the reader.

We have followed reviwer#1 and corrected the way that abbreviations were presented. We also included a list of abbreviations on page 19.

  • Instead of Mg ++ or Ca ++, I think it should be Ca2+ (with superscript); also, please pay attention to the use of subscript and superscript throughout the manuscript.

Following this advice, we have changed the abbreviation for the ions. We have reviewed all the superscripts in the text too.

  • The figures would benefit from captions explaning the use of italic  and even the explanation of some of the abbreviations used.

As reviewer#1 has suggested, we have added a more detailed explanation for the use of italics and abbreviations in figures, especially for Figure 1 (please, see page 3).

  • please separate table ticle from its footnotes which should contain the abbreviations used and other explanations.

We agree with Reviewer#1. We have changed the position for the abbreviation in the tables (please, see pages 10 and 11) since we believe that this action will clarify the understanding of the information.

  • I believe that you should re-write the Conclusions section to better underline the point of the paper and highlight its strenghts.

We thank reviewer#1 for improving the editing of the manuscript. We have followed your suggestions and corrected the conclusion making it better to understand. We have focused on the most relevant points in each topic: diagnosis, prognosis and treatment of the diseases. We firmly believe that we have created a more direct take-home message.

Reviewer 2 Report

It is a review of the Bartter and Gitelman Syndromes very interesting, very well written and very complete. Only minor considerations:

-In section 3.2 of Molecular basis of Bartter syndrome and clinical consequences, I think it would be positive to comment   globally that in terms of presentation there is a neonatal , that may have antenatal manifestations and classic Bartter syndrome, which can appear at any time in life.

-I think it would be of interest to make a reference to the prognosis, for example in section 4.1.6, specifically reflecting it

-On page 5, lines 143-144 you write about a recent study, but it is not recent because it is from 2013

-it is important to emphasize that for pregnant women with early onset and severe polyhydramnios without morphological anomalies, antenatal Bartter syndrome should be highly suspected.

Author Response

We are pleased to see that the reviewer was generally enthusiastic about our article, though he/she has requested to add more information about some aspects. We agree that including this information would strengthen the manuscript and thus we have performed all the requested suggestions and corrections. We also wish to thank the reviewer for his/her helpful comments and suggestions to improve our manuscript.

Please find our “point-by-point” responses to the specific questions raised by the reviewer#2. 

  • In section 3.2 of Molecular basis of Bartter syndrome and clinical consequences, I think it would be positive to comment globally that in terms of presentation there is a neonatal , that may have antenatal manifestations and classic Bartter syndrome, which can appear at any time in life.

We have followed the reviewer’s advice since neonatal and classic Bartter presentation is the most common classification, as well as an easy categorization. Thus, we have included this idea on page 5 (lines from 159 to 169).

  • I think it would be of interest to make a reference to the prognosis, for example in section 4.1.6, specifically reflecting it

We thank reviewer#2 for improving the editing of the manuscript. Therefore, we have added specific lines related to the prognosis in Gitelman and Bartter syndromes within the aforementioned section 4.1.6. Specifically, prognosis is referred on page 18, from 607 to 617 lines.

  • On page 5, lines 143-144 you write about a recent study, but it is not recent because it is from 2013.

We apologised for our mistake. In consequence, we have corrected this error in the aforementioned page.

  • It is important to emphasize that for pregnant women with early onset and severe polyhydramnios without morphological anomalies, antenatal Bartter syndrome should be highly suspected.

We totally agree with this statement. Hence, we highlight this idea on page 13 (lines from 372-374).

Reviewer 3 Report

This review is adequate for publication. Its is comprehensive and accademical. Only there are several tipographical errors which should be accuratelly corrected

Author Response

This review is adequate for publication. Its is comprehensive and accademical. Only there are several tipographical errors which should be accuratelly corrected

We are pleased to see that the reviewer was generally enthusiastic about our review, though he/she has requested to add more information about some aspects. We agree that including this information would strengthen the manuscript and thus we have performed all the requested suggestions and corrections. We also wish to thank the reviewer for his/her helpful comments and suggestions to improve our manuscript.

As reviewer#3 has mentioned in his/her comments, we totally agree with him/her. Then, we have corrected all the typographical mistakes including in the text. Moreover, all the text was completely proofread by a native English speaker to improve the text
